# Integrative Spatial Proteomics and Single-Cell RNA Sequencing Unveil Molecular Complexity in Rheumatoid Arthritis for Novel Therapeutic Targeting

**DOI:** 10.3390/proteomes13020017

**Published:** 2025-05-22

**Authors:** Xue Wang, Fei Wang, Archana S. Iyer, Heather Knight, Lori J. Duggan, Yingli Yang, Liang Jin, Baoliang Cui, Yupeng He, Jan Schejbal, Lucy A. Phillips, Bohdan P. Harvey, Sílvia Sisó, Yu Tian

**Affiliations:** 1AbbVie, South San Francisco, CA 94080, USA; xue.wang@abbvie.com; 2AbbVie Bioresearch Center, Worcester, MA 01605, USA; fei.wang1@abbvie.com (F.W.); archana.iyer@abbvie.com (A.S.I.); heather.knight@abbvie.com (H.K.); lori.duggan@abbvie.com (L.J.D.); yingli.yang@abbvie.com (Y.Y.); liang.jin@abbvie.com (L.J.); baoliang.cui@abbvie.com (B.C.); jan.schejbal@abbvie.com (J.S.); lucy.phillips@abbvie.com (L.A.P.); 3AbbVie, North Chicago, IL 60064, USA; yupeng.he@abbvie.com; 4AbbVie, Cambridge, MA 02139, USA; bohdan.harvey@abbvie.com

**Keywords:** rheumatoid arthritis, laser capture microdissection, untargeted proteomics, mass spectrometry, multi-omics integration, scRNA-seq, membrane proteins, extracellular matrix

## Abstract

Understanding the heterogeneity of Rheumatoid Arthritis (RA) and identifying therapeutic targets remain challenging using traditional bulk transcriptomics alone, as it lacks the spatial and protein-level resolution needed to fully capture disease and tissue complexities. In this study, we applied Laser Capture Microdissection (LCM) coupled with mass spectrometry-based proteomics to analyze histopathological niches of the RA synovium, enabling the identification of protein expression profiles of the diseased synovial lining and sublining microenvironments compared to their healthy counterparts. In this respect, key pathogenetic RA proteins like membrane proteins (TYROBP, AOC3, SLC16A3, TCIRG1, and NCEH1), and extracellular matrix (ECM) proteins (PLOD2, OGN, and LUM) showed different expression patterns in diseased synovium compartments. To enhance our understanding of cellular dynamics within the dissected regions, we further integrated the proteomic dataset with single-cell RNA sequencing (scRNA-seq), and deduced cell type enrichment, including T cells, fibroblasts, NK cells, myeloid cells, B cells, and synovial endothelial cells. By combining high-resolution spatial proteomics and transcriptomic analyses, we provide novel insights into the molecular mechanisms driving RA, and highlight potential protein targets for therapeutic intervention. This integrative approach offers a more comprehensive view of RA synovial pathology, and mitigates the limitations of traditional bulk transcriptomics in target discovery.

## 1. Introduction

Rheumatoid arthritis (RA) is a chronic autoimmune disease characterized by persistent inflammation, fibroblast proliferation, and progressive joint damage. The complexity of RA is highlighted by its diverse molecular and cellular landscape. Recent advancements in single-cell RNA sequencing (scRNA-seq) have uncovered substantial heterogeneity among the lymphocyte, monocyte, endothelial cell, and fibroblast populations within the synovium of affected individuals [1,2]. This heterogeneity poses significant challenges in RA diagnosis and its treatment, which is not universally effective, and calls for the identification of novel biomarkers and druggable targets to improve patient outcomes.

While scRNA-seq provides a high-resolution of dynamics of gene expression at the single-cell level, it lacks the topographical or spatial context, therefore could not enable the correlation of gene expression profiles with microscopically resolved tissue regions. To overcome this limitation, both spatial transcriptomic and proteomic technological advances in the last decade have successfully captured comprehensive transcriptome and proteome information from histological sections. For example, laser capture microdissection (LCM) coupled with ultra-sensitive shotgun bottom-up liquid chromatography–mass spectrometry (LC-MS) or LCM-spatial proteomics, is a powerful technology that allows us to analyze the proteome with precise spatial resolution [3,4]. Unlike traditional proteomics, which often relies on bulk tissue samples, LCM-spatial proteomics allows for the targeted isolation of specific cellular regions from complex tissues, and preserves contextual information regarding protein expression within the tissue architecture. This capability is particularly critical in studying heterogeneous diseases, where understanding the spatial context of protein abundance of different histopathological features can provide valuable insights into disease pathogenesis [4,5,6]. Similarly, the spatial proteomics fingerprints embedded in multicellular domains can inform on the status of the ECM remodeling and complex cellular cross talking (cell-to-cell interactions based on receptor to ligand contacts or on locally secreted factors), profiling of pathogenetic niches further underlying specific tissue biomarkers and disease-modifying therapeutic targets [7,8]. Like LCM-proteomics, novel protein-based techniques such as NanoString GeoMx Digital Spatial Profiling (DSP) do not achieve single-cell resolution, but enable high-plex protein analysis (~600 targeted proteins) from FFPE tissue sections [9]. On the other hand, mass spectrometry-based techniques like Matrix-Assisted Laser Desorption/Ionization Mass Spectrometry (MALDI-MS) allows detailed mapping of protein distribution across tissue sections, but the large set of protein identifications and quantifications are still challenging, especially for low-abundance proteins [10]. Some other MS techniques can provide single-cell resolution, but with limited protein coverage [11].

In this study, we precisely selected and subsequently isolated specific regions of interest (ROI) of synovium from both RA patients and normal healthy (NH) subject controls using laser-mediated capture. This approach allowed us to capture the key spatially resolved histopathological features of the RA synovium like the hyperplastic lining, the immune-rich superficial sublining, and the proliferation of the fibrous-rich deeper sublining. Additionally, we combined spatial proteomics data with published scRNA-seq data to achieve spatially resolved cellular characterization. By isolating ROIs from RA and NH synovial samples, and analyzing their protein content, we uncovered several membrane proteins, and extracellular matrix (ECM) and matrisome proteins involved in the pathophysiology of RA. This work successfully offers insights into potential biomarkers and targets for RA diagnosis and therapy, respectively, and paves the way for the development of targeted therapeutic strategies.

## 2. Materials and Methods

### 2.1. Study Subjects’ Information

Ethically sourced synovial specimens from healthy joints and from subjects with rheumatoid arthritis (RA) were procured from various biorepository sites like National Disease Research Interchange (NDRI), Discovery Life Sciences and Avaden Biosciences. All collections and processing related to human specimens were performed in accordance with the relevant guidelines and regulations, and complying with de-identification practices and ethical donor authorization. Upon receipt, a total of nine formalin-fixed paraffin-embedded synovial blocks were selected based on mRNA quality, tissue architecture, and cellular morphology. Three synovial samples with no evidence of histopathology were included as normal-healthy (NH) samples and six as RA, based on containing histopathological hallmarks of the disease. The average age of the normal-healthy (NH) controls and RA patients were 75.3 ± 20.5 and 58.6 ± 15.0 years, respectively. The cause of death for the three NH was head trauma (NH1), dementia (NH2), and cardiorespiratory arrest (NH3). Demographic data on the sex, age, joint sampled, and microscopic pathotype is provided in Table 1.

### 2.2. Histology, Immunohistochemistry, and In Situ Hybridization

All formalin-fixed paraffin-embedded (FFPE) blocks were sectioned at 5 µm in Diethyl pyrocarbonate (DEPC)-treated water, and stained with hematoxylin-eosin (H&E). Additional sections were processed for immunohistochemistry (IHC) and screened for mRNA quality by in situ hybridization (ISH) using the positive control probe PPIB (Cat # 313908) and the negative control probe, dapB (Cat # 312038). Immunohistochemistry using the Leica Bond RX automated immunostainer (Leica Biosystems, Deer Park, IL, USA) was performed to assess spatial microenvironments rich in T cells, B cells, macrophages, and fibroblasts. Prior to loading on the Bond RX, synovial sections were baked for 60 min at 60 °C. Following a dewax step, slides were pretreated with either EDTA or citrate heat-induced epitope retrieval (Leica). Slides were blocked with both a dual endogenous enzyme block and protein block (both from Agilent, Santa Clara, CA, USA). They were incubated with the primary antibodies (Appendix A), detected with either horseradish peroxidase or alkaline phosphatase polymer, and visualized with either 3, 3′-diaminobenzidine (DAB, Leica Bond Polymer Refine Detection kit, Leica Biosystems, Deer Park, IL, USA) or red chromogen (Leica Bond Polymer Refine Red Detection kit, Leica Biosystems, Deer Park, IL, USA).

### 2.3. LCM Procurement and Tissue Section Preparation

On day 1, six serial sections, each 5 microns thick, were cut per block. The first section was mounted on a regular glass slide (Surgipath Apex Superior adhesive slides, Leica Biosystems, Lincolnshire, IL, USA; Cat #3800080) and stained with hematoxylin and eosin (MilliporeSigma, Burlington, MA, USA; Cat #GHS1128-4L). The next five serial sections were mounted on PEN Membrane slides (Leica Microsystems Inc, Deerfield, IL, USA; Cat #11505189) and stained with Meyer’s hematoxylin for 30 s, then held at 4 °C until day 3. The H&E slide was digitally imaged using a Pannoramic 1000 whole slide scanner at 20× magnification (3DHISTECH, Budapest, Hungary). On day 2, the image was imported into Visiopharm (Version: 2024.07.1.16912 x64) and ROI were annotated by a pathologist. ROI included lining, areas of fibrous sublining, and areas of immune sublining. On day 3, the annotated H&E images were used to guide the manual LCM capture on the hematoxylin-stained PEN membrane slides. For each ROI, ~2 mm^2^ of tissue was captured into 96-well plate (Eppendorf, Enfield, CT, USA; Cat #0030129512) and placed at −80 °C. According to recent improvements of our LCM proteomics platform, a lower LCM size like 0.5 mm^2^ was feasible without compromised protein IDs (Appendix A).

Before preparation for proteomic analysis, the plate was centrifuged at 2000× *g* for 5 min to aggregate LCM cuts. The sample preparation followed a previously published method [4]. Briefly, 20 µL of 20 mM ammonium bicarbonate was added to each sample well. The plate was sealed with adhesive aluminum film (Corning, Corning, NY, USA; Cat #CLS6570), and shaken in a Thermomixer at 600 rpm for 1 min. After that, the plate was centrifuged again for another 5 min and then heated at 100 °C for 70 min with the Thermomixer at 300 rpm. After cooling down the plate completely, 5 µL buffer containing 60% acetonitrile with 100 mM ammonium bicarbonate were added for another 30 min heating at 75 °C. For protein digestion, 1 µL of 0.5 µg/µL LysC (1:10, enzyme: protein) was added to the room-temperature plate for 3 h incubation at 37 °C. And then another 1 µL of 0.5 µg/µL Trypsin was added for overnight digestion at 37 °C. To terminate the digestion, 1 µL 30% trifluoroacetic acid (TFA) was added to reach a final TFA of 1%. The dried peptides were resuspended with 18 µL LC injection buffer (2% MeOH, 0.1% TFA in water). Detailed LCM sample metadata including area size, peptide yield, and concentration are listed in Appendix A.

### 2.4. Liquid Chromatography–Mass Spectrometry (LC-MS)

The 300 ng peptides of each sample were analyzed using a nanoElute UHPLC coupled to timsTOF Pro 2 Mass Spectrometer (Bruker Daltonics, Bremen, Germany) with parallel accumulation serial fragmentation (PASEF). The HPLC separation (Solvent A: 0.1% formic acid in water; Solvent B: 0.1% formic acid in acetonitrile) was carried out with an Aurora ultimate CSI C18 column (75 µm × 25 cm, 1.7 µm, IonOpticks, Notting Hill, Australia) heated at 50 °C with a 30 min gradient (5–9% B for 4 min, 9–36% B for 21.5 min, 36–95% for 0.5 min and 95% for 4 min to wash the column; flowrate at 0.36 μL/min). Mass-spectrometric data were acquired in DIA-PASEF positive mode with scanning range of 100–1700 *m*/*z*. The window settings followed mass width of 26.0 Da, 32 mass steps per cycle covering the mass range from 400 to 1201 *m*/*z* and 1.79 s of estimated cycle time.

### 2.5. Proteomics Data Analysis

A database search of raw mass spectrometry files was performed using DIA-NN (version1.8.1) [12] with an in silico library based on Uniprot human protein database (downloaded on 12 January 2021). The search employed the default settings with Trypsin/P protease, peptide length range of 7–30, precursor charge range of 1–4, precursor range from 300 to 1800 *m*/*z*, and fragment ion *m*/*z* range from 200 to 1800. RT-dependent cross-run normalization was used, and the protein names from FASTA were defined as protein inference. All search was conducted with match-between-run (MBR) off. Quantified protein group IDs and peptide IDs are summarized in Appendix A. After the quantitative dataset was generated, the following analysis was performed in the R framework (version 4.1.2). The protein abundance data (report-first-pass.pg_matrix) was filtered to ensure that each protein group contained <50% missing values or <1 missing value in any ROI. The protein abundances were then log2 transformed and normalized using “medianNormalization” function in the “NormalyzerDE” R package (version 1.16.0). Remaining missing values were imputed using “missForest” function in the “missForest” R package (version 1.4) [13,14]. Differential expression (DE) analysis was performed using a linear model-based framework implemented in the “limma” R package (version 3.50.0) [15]. A Benjamini–Hochberg adjusted *p* value (false discovery rate or FDR) of less than 0.05 and a greater than 2-fold change (FC) was considered statistically significant. The enriched activated and suppressed pathways in pairwise protein expression comparisons (Log_2_ FC) were determined by gene set enrichment assay (GSEA) using the “WebGestaltR” R package against the Gene Ontology (GO) terms of biological processes [16,17].

### 2.6. Skyline Visualization of MS Spectra

To assess the quality of spectra manually, we employed Skyline (version 23.1.0.455) to visualize the outputs from DIA-NN, including quantification results and the spectral library. We initiated a new project centered on DIA spectral library searches, and imported the DIA-NN-generated spectral library (.speclib) with a default q-value threshold of 0.05. Subsequently, raw files from timsTOF were imported to extract chromatograms, incorporating oxidation modifications. The transition settings utilized included precursor charges of 2 and 3, ion charges of 1 and 2, and ion types y, b, and p, with an ion match tolerance of 0.05 *m*/*z*. Full-scan settings adhered to the optimal configurations from the Skyline DIA tutorial, featuring a centroided mass analyzer for precursor/product masses and a mass accuracy of 20 ppm. Human FASTA files (.fasta) sourced from UniProt were also loaded into Skyline. This setup facilitated the matching of 6618 protein groups, covering 33,346 peptides, 73,396 precursors, and 660,564 transitions, with each group containing at least one peptide from a representative raw file. Additionally, we imported the DIA-NN quantitative report file (.tsv) to define peak boundaries more accurately. Utilizing this configuration, we harnessed Skyline’s effective visualization capabilities to examine chromatograms and spectral data, focusing on precursor and product ion chromatograms, to evaluate experimental spectra against the library data for confirming peptide identifications, thus providing a thorough analysis within a unified Skyline’s analytical framework. For clarification, all of the other bioinformatics analyses and data mining are based on the DIA-NN output protein group intensity files. Skyline analysis is mainly for spectral data quality control.

### 2.7. Integration of LCM Proteomics and scRNA Dataset

An unpaired two-tailed Student’s *t*-test (*p* < 10^−2^, calculate effect size with Cohen’s d value) was used to identify differentially expressed protein features between the target region of interest and all other non-target regions in the LCM spatial proteomics dataset. For each gene with *p* < 10^−2^, we examined in which region the effect size was highest, and determined that protein to be specific or uniquely up-regulated for this region. This method to identify region-specific proteins is referred from a multi-omics integration analysis paper in pancreas cancer [18]. We have also utilized this method to identify cell type enrichments in pathological regions in idiopathic pulmonary fibrosis by combining LCM spatial proteomics and scRNAseq transcriptomics [19]. The whole integration method used in the current paper is described in detail in a previous report [19]. Here, we extracted 68 up-regulated region-specific differential proteins for NH lining, 269 for NH sublining, 970 for RA lining, 82 for RA fibrous sublining, and 453 for RA-immune sublining (Appendix A). Next, we integrated with the scRNA-seq dataset by querying z-scores of up-regulated region-specific differentially expressed protein-corresponding genes in distinct cell types. We referred to it as the Query method, as previously described [19].

The scRNA-seq transcriptomics data were referred from paper published by Zhang and co-authors [1]. They performed multi-modal scRNA-seq and multicolor immunofluorescence staining in synovial membrane from RA (*n* = 70) and Osteoarthritis (OA) (*n* = 9) donors to build single-cell atlas that include more than 314,000 cells and identified 58 proteins and 30,997 genes with 15 common protein/gene pairs after canonical correlation analysis and batch effect correction. We combined expression profiling from 58 proteins and 30,982 unique genes, and obtained 31,040 features. In total, 99.3% (3790 of 3816) protein corresponding genes from the LCM spatial proteomics dataset were also listed in these 31,040 features. Further, 77 subclusters or distinct cell states in RA and OA were annotated in six cell types: 9 B/plasma clusters, 5 Endothelial clusters, 15 Myeloid clusters, 14 NK cell clusters, 10 Stromal clusters, and 24 T cell clusters. The average expression level of 31,040 features in each of 77 clusters were extracted from Accelerating Medicines Partnership^®^ (AMP) RA phase II database: https://immunogenomics.io/ampra2/ (accessed on 13 September 2024).

The average expression level of 31,040 features from all cells in each of 77 clusters were extracted from the AMP RA phase II database (https://immunogenomics.io/ampra2/, accessed on 13 September 2024) and normalized (z = (X − μ)/σ) further in R. The previous section identified up-regulated region-specific differential proteins from the LCM spatial proteomics dataset. Then, the relative average expression levels of up-regulated region-specific differential protein corresponding genes in a specific cluster were summed and divided by the region-specific protein set number. The resultant number is the z-score for that region in that cluster. Following this rule, a paired region/cluster association z-score matrix was produced. Positive values indicate enrichment of the cluster in the histopathological region type, while negative values indicate depletion. The absolute values quantify the extent.

## 3. Results

### 3.1. Subjects and Synovium ROI Selections

A trained pathologist confirmed optimal tissue architecture and cellular morphology of H&E-stained tissue sections. All NH synovial samples were diagnosed as non-lesional based on the lack of lesions in both synovial lining (SL) and synovial sublining (SSL). For the RA counterpart, samples were classified into pathotypes [20] based on histopathological examination and ancillary immunohistochemistry. Recent works describing RA pathotypes uncovered three distinct synovial pathotypes: (i) cellular dense, lymphocyte rich (lymphoid), (ii) myeloid rich with few lymphocytes (diffuse/myeloid), and (iii) fibroblast rich (pauci-immune), which are identifiable through distinct cellular and tissue level changes within synovial joint biopsies. Accordingly, all RA samples classified with the lymphomyeloid pathotype contained multifocal clusters of CD19^+^ lymphocytes with morphology consistent with B cells and admixed numbers of CD3^+^ T cells, all surrounded by large numbers of plasma cells consistently located along the superficial sublining. Smaller numbers of CD3^+^ T cells were also detected in deeper areas of the sublining neighboring blood vessels. Immunohistochemical detection of CD68^+^ macrophages spanned all compartments of the synovium. Representative H&E and CD19, CD3, and CD68 immunohistochemical images from a sample of the lymphomyeloid pathotype are shown in Figure 1B (top panel). One of the RA samples, RA2, had a single lymphoid aggregate with minimal numbers of CD20^+^ cells, a few T cells, and frequent macrophages throughout the superficial synovial sublining. This sample with poor B-cell representation was classified as diffuse myeloid, following previously established criteria [20,21].

To prepare LCM dissection, a pathologist manually annotated H&E stained 2 mm^2^ ROIs encompassing the three major spatially resolved regions represented in the RA synovium (the SL, the immune-rich SSL, and the fibrous-rich SSL; Figure 1A). The NH SL was 0–3-cell-thick layer, whereas the RA synovium lining varied from 3 to 10 cells in thickness, depending on the ROI. The SSL in NH subjects appeared to be composed of loose connective tissue and adipose tissue. For most RA subjects, the SSL was further annotated according to areas (i) that were highly infiltrated with immune cells forming aggregates (immune-rich sublining, or immune SSL), and (ii) that were rich in fibrovascular stroma (fibrous-rich sublining, or fibrous SSL) (Figure 1A). In RA subjects, immunohistochemistry for the detection of immune cells and fibroblast aided the selection of immune rich and fibrous-rich areas, respectively (Figure 1B). All three healthy samples and one RA sample (RA2) had minimal to low numbers of resident CD3^+^ and CD19^+^ lymphocytes. The RA2 sample had moderate numbers of CD68^+^ macrophages, though. The remaining RA samples (RA1, RA3-RA6) displayed high synovial infiltration of lymphocytes and macrophages. In RA, most immune cells were consistently distributed focally to diffusely along the superficial SSL with low counts surrounding blood vessels of the deeper layers of the SSL (Figure 1B). In addition, CD68^+^ macrophages were notably abundant in the synovial lining and in the deep SSL. Fibroblast markers like CD248, FAP, PDPN, and CD90 (Figure 1B) were well represented in the synovium of RA subjects. CD248^+^ and FAP^+^ fibroblasts were most abundant in both lining and deep fibrous-rich SSL, whereas CD90^+^ fibroblasts were localized to the SSL in association with blood vessels and PDPN+ fibroblasts were most abundant in the SL and its immediate SSL.

The minimum requirement of 2 mm^2^ ROIs was achieved for the SL in all NH and RA subjects. For NH1-NH3, RA1, and RA2, the SSL ROI included both superficial and deep SSL as a single ROI since the immune-rich SSL was not prominent (NH samples) or due to the small size of the sample (RA1 and RA2). However, for RA3-RA6, 2 mm^2^ ROIs were successfully dissected from the SL, the immune-rich SSL, and the fibrous-rich SSL (Figure 1C). During microdissection, all required 2 mm^2^ ROI were completed for each subject using up-to-two available serial hematoxylin-stained tissue sections (Figure 1C,D). The hematoxylin stain performed on the membrane slide allowed for the identification of ROI morphology in serial sections.

### 3.2. Protein Quantifications of Synovium Using LCM Coupled with LC-MS

Using the in-house spatial proteomics workflow (Figure 2A), we generated a protein expression abundance dataset containing 3818 quantified proteins after processing through clean-up, log2 transformation, filtering, and missing value imputation (see Methods for details). To explore clustering within histopathological regions and disease states, we performed Principal Component Analysis (PCA) (Figure 2B). Clear separations were observed along the principal component 1 (PC1), particularly between NH and RA groups in both the lining and sublining compartments (Figure 2C,D), highlighting substantial differences in protein expression profiles across disease conditions. Within the NH groups, a distinct clustering emerged between the SL and SSL compartments (Figure 2E), reflecting unique protein expression characteristics of different synovial compartments.

In the RA groups, a minor overlap was seen between the SL and fibrous SSL compartments (Figure 2F), suggesting some shared features; however, overall separation along PC1 indicates significant differences between these two compartments. Additionally, only a small portion of distinct clustering was noted between immune SSL and SL compartments within the RA groups (Figure 2F), suggesting similar protein profiles based on the primary components analyzed. We further conducted differential expression (DE) analysis to identify significantly altered proteins (|FC| > 2, FDR < 0.05) across disease states (Figure 2G) and tissue compartments (Figure 2H). Results showed that protein expression changes were more pronounced between disease conditions (Figure 2H) than between different histopathological compartments (Figure 2G), which are consistent with PCA plotting results (Figure 2B). Moreover, fewer protein changes were observed among different tissue compartments within RA synovium compared to healthy synovium, possibly due to relatively large variances caused by more samples involved in RA group (Figure 2G). Moreover, we performed PCA in only two target groups (Appendix A) and compared output PCA loadings in the first two principal components, PC1 and PC2, with corresponding Log_2_ FC in differential expression analysis (Figure 2G,H), and they showed correlations (Appendix A). Proteins with maximum loadings make highest contributions to separate on each PC. They are also the most differentially expressed proteins, especially in comparisons with clear PCA separation (Appendix A). These strong Pearson correlations validate our methods.

### 3.3. Membrane Proteins and Matrisome Profiling of Different Synovium Regions

A Gene Set Enrichment Analysis (GSEA) against Gene Oncology terms of Biological Process using pairwise protein expression comparisons (Log_2_ FC) among different histopathological regions was conducted in this study, and it identified the activation of multiple membrane protein-related processes and extracellular structure organization, especially in RA fibrous SSL (Figure 3A–C and Appendix A). Therefore, we focused on these two categories of proteins: (1) membrane proteins, and (2) extracellular matrix (ECM) or matrisome proteins.

Herein, 60 of 411 membrane proteins (57 up-regulated and 3 down-regulated) with significant differences between RA and NH groups were uncovered (both appear in RA SL versus NH SL, and in RA fibrous SSL versus NH SSL comparisons) (Figure 3D). These proteins included 18 transmembrane receptors, two phosphatases, seven transporters, one kinase, eight enzymes, one G-protein coupled receptor, three ion channel proteins, and 20 other proteins based on QIAGEN Ingenuity Pathway Analysis (IPA) annotation [22]. Moreover, within 178 matrisome proteins annotated by reference [23], 24 significantly changed matrisome proteins in RA samples (18 up-regulated and 6 down-regulated, both appear in RA SL versus NH SL and RA fibrous SSL versus NH SSL comparisons), including 6 core matrisome proteins and 18 matrisome-associated proteins (Figure 3E). The six core matrisome proteins consist of ECM glycoproteins (DPT, MFAP1, CILP, SPARC and LRG1) and proteoglycan (OGN), while the 18 matrisome-associated proteins contain secreted factors (S100A9, S100A8, IL17B, S100A11), ECM regulators (SERPINH1, PLOD3, PLOD1, CTSZ, CTSA, PLOD2, CTSS, SERPINA3, AMBP, CTSD, CSTB, SERPINB8, and ELANE) and ECM-affiliated protein (LMAN1).

Among 60 membrane proteins and 24 matrisome proteins significantly changed in RA samples, in this study, we were focused on proteins with significant difference between RA SL and fibrous SSL or immune SSL considering the heterogeneity of RA tissue. The three membrane proteins (TYROBP, ACO3 and SLC16A3) showed significant changes in both comparison of RA SL vs. RA fibrous SSL and comparison of RA SL vs. RA-immune SSL. TYRO protein tyrosine kinase-binding protein (TYROBP), also known as DAP 12, a transmembrane adaptor protein involved in immune cell signaling [24], showed an elevated expression in RA synovium including SL, fibrous and immune SSL regions, and the RA SL expressed higher level of TYROBP than other SSL areas, which could contribute to the inflammation and joint destruction of RA disease (Figure 4A). Amine Oxidase Copper-containing 3 (AOC3), also known as VAP-1, is a dual-function glycoprotein with both adhesive and enzymatic properties, which expresses on the surface of endothelial cells, adipocytes and some immune cells [25]. Here, we demonstrated higher levels of AOC3 in SSL including both fibrous and immune SSL than in SL, regardless of disease status (Figure 4B), which further supported the heterogenous distribution of AOC3 in synovium. When comparing the disease conditions, NH samples show higher levels of AOC3 than RA samples, regardless of histopathological region type (Figure 4B). SLC16A3 solute carrier family 16 member 3 (SLC16A3), also known as monocarboxylate transporter 4 (MCT4), is implicated in the pathophysiology of RA through maintaining the glycolytic metabolism of synovial fibroblasts and immune cells [26]. The elevated SLC16A3 expression in RA synovium we found here may contribute to RA pathogenesis through mediating metabolic microenvironment and joint inflammation (Figure 4C). Targeting SLC16A3 may offer a novel therapeutic approach to modify the metabolic and inflammatory environment in RA. Additionally, we found two other membrane proteins (TCIRG1 and NCEH1) that are significantly upregulated in RA SL region compared to NH SL and RA-immune SSL. T cell immune regulator 1 (TCIRG1) is related to osteoclast-mediated bone resorption during RA disease development [27,28]. The increased TCIRG1 we detected is a result of activated osteoclasts in RA, which contribute to bone erosion (Figure 4D). Targeting TCIRG1 may provide a strategy to mitigate bone loss and joint destruction in RA. The role of neutral cholesterol ester hydrolase 1 (NCEH1), or KIAA1363, is an emerging research area that connects lipid metabolism with the inflammatory processes of the RA disease [29]. The elevated NCEH1 of RA synovium in this study correlated to the chronic inflammation (Figure 4E). Targeting NCEH1 provides a potential therapeutic avenue to alleviate inflammation and joint destruction in RA.

When the 24 matrisome proteins were further explored, we found three proteins with significant difference among different RA areas. Procollagen-Lysine, 2-Oxoglutarate 5-Dioxygenase 2 (PLOD2) is an enzyme crucial for collagen cross-linking and ECM stabilization [30]. Here, we found an elevated expression of PLOD2 in RA SL compared to fibrous SSL, immune SSL, and NH SL (Figure 4F), which might contribute to RA tissue fibrosis, synovial hyperplasia, and chronic inflammation. Targeting PLOD2 would manage ECM remodeling and reduce joint damage in RA. Another detected protein critical in ECM remodeling is Osteoglycin (OGN), also known as Mimecan, a small leucine-rich proteoglycan (SLRP) [31]. Our results showed the downregulated OGN in RA SL and fibrous SSL compared to NH group (Figure 4G), and its expression in RA SL and RA-immune SSL, were significantly lower than fibrous SSL in RA synovium, which contributes to RA pathogenesis through dysregulated ECM structure organization. Similarly, Lumican (LUM), a member of SLRP family related to collagen fibrillogenesis and ECM organization [32,33], downregulated in RA fibrous and immune SSL compared to NH SSL (Figure 4H). Within RA synovium, LUM expressed higher level in fibrous SSL than SL and immune SSL (Figure 4H). The downregulation in lumican levels is often associated with disorganization of the extracellular matrix environment that exacerbates joint inflammation and degradation, which provides the value as a therapeutic target [32,34].

### 3.4. Cell Type Enrichment Through Integration with scRNA Sequencing

To uncover the cell types enriched in each of the five histopathological microenvironments (NH SL, NH SSL, RA SL, RA-immune SSL, and fibrous SSL), an integration analysis combining our LCM proteomics dataset with a published scRNA sequencing dataset (see Section 2) was executed by a Query method, which utilized a z-score to evaluate positive correlations between histopathological regions and cell clusters. We extracted up-regulated, region-specific differentially expressed proteins from each of the five distinct histopathological region types in our LCM proteomics dataset, and queried, then summed, the average expression of these region-specific protein-corresponding genes as z-scores in each of cell subclusters from the scRNA-seq dataset. A positive z-score means an enriched expression of up-regulated region-specific proteins in a cell subcluster, while a negative z-score means a depleted expression. The absolute value indicates the quantified extent of enrichment or depletion. For clear visualization, the enrichment (Figure 5) and depletion (Appendix A) matrix maps are separately displayed. If a subcluster is not enriched or depleted in any of the five microenvironments, then it is omitted from the enrichment maps (Figure 5 and Appendix A).

For the 24 T cell subclusters analyzed in this study, the CD4^+^ and CD8^+^ cell subsets are mostly enriched in RA-immune SSL-like CD4^+^ T_fh_ and T_ph_ cells, regulatory CD4+ T cells (T_reg_, T-8), CD146+ memory cells (T-11), CD8^+^ granzyme K/B+ memory cells (T-13), and CD8^+^ activated/NK-like cells (T-17) (Figure 5A). The CD38^+^ T cells (T-20) and γδ Vδ1 T cells (T-22) are also exclusively expressed in the immune SSL of RA synovium (Figure 5A).

The lining fibroblasts were divided into PRG4^+^ CLIC5^+^ (F-0), PRG4^+^ (F-1), and RSPO3^+^ (F-8) subsets, and were identified with more enrichment in NH SL compared with RA SL (Figure 5B). This is supportive of similar observations by others in that the homeostatic SL decreases with RA progression [35,36]. The sublining fibroblasts with HLA-DRA^+^ (F-5 and F-6), CD34^+^ (F-2), DKK3^+^ (F-4), POSTN^+^ (F-3), and NOTCH3^+^ (F-7) populations were enriched more in the RA fibrous SSL region (Figure 5B). Among them, the CD34^+^ cluster (F2) is associated with differentiation and progenitor-like status, the inflammatory CD74hi HLAhi cluster (F-5) and a CXCL12^+^ SFRP1^+^ cluster (F-6) highly express IL16, which has been a known RA druggable target.

The 14 clusters of innate lymphocytes were also captured in our study, the exclusive enrichments of CD56^low^CD16^+^ and CD56^hi^CD16^−^ population were discovered in the immune SSL of RA synovium (Figure 5C). The CD56^low^CD16^+^ population comprises IFNG- (NK-0), IFNG^+^CD160^+^ (NK-1), IFNG^+^CD160^−^ (NK-2), and GZMB^−^ (NK-3) subtypes, while CD56^hi^CD16^−^ population includes GNLY^+^CD69^+^ (NK-7) and IFN response (NK-8) cells. PCNA^+^ (NK-10) and MKI67^+^ (NK11) proliferating cells also show strong enrichment in the RA-immune SSL region compared with all others (Figure 5C). After integrating the 15 myeloid clusters with regional protein readouts, four MERTK^+^ macrophage clusters (M-0, M-1, M-2, and M-3) were identified widely in synovium tissue with more enrichment in RA lining (Figure 5D). Similar expression trends were also found in the SPP1^+^ (M-4) and STAT1^+^ CXCL10^+^ (M-6) clusters. The dendritic cell populations (M-12, M-13) enriched more in RA SL and immune SSL regions (Figure 5D). The synovial endothelial cells were divided into lymphatic (E-4) and blood subsets (SPARC^+^, LIFR^+^, ICAM1^+^, NOTCH4^+^). These cell subsets showed enrichment among different compartments of synovium (Figure 5E). For the analysis of six B cell clusters and three plasma cell clusters, only AICDA^+^BCL6^+^ GC-like B cells were enriched in all three synovium tissue components with a higher expression in RA-immune SSL (Figure 5F). Another B cell type of CD11c^+^ LAMP^+^ ABC showed exclusive enrichment in RA-immune SSL (Figure 5F).

### 3.5. Visual Verifications of Key Fibroblast Proteins Identified by Spatial LCM-Proteomics with Its IHC Counterpart Spatially Identified in Tissue Sections

All synovial samples included in this proteomics study have been previously evaluated for various fibroblast protein markers. CD90 (THY-1), a glycoprotein expressed on the surface of fibroblasts, neurons, and some T cells, was reported to be related to immune modulation and fibroblast activity serving as a potential RA biomarker [37]. The expression of CD90 was detected with mass-spec, and increased in RA in the SL and fibrous as well as immune SSL (Figure 6A). Spatial evaluation of immunohistochemically detected CD90 confirmed increased detection in RA SSL, mostly in association with blood vessels (Figure 6E,I). No CD90 was detected in the SL of the synovium. Another remarkable difference between RA and NH synovium was with fibroblast markers like CD248 and FAP. CD248 (endosialin) is a cell surface glycoprotein involved in various cellular processes, including cell adhesion, migration, and tissue remodeling related to RA pathophysiology [38]. Immunohistochemical detection of CD248^+^ fibroblasts confirmed and increase in RA SL and SSL fibroblasts compared to NH synovium (Figure 6F,J). In the NH synovium, while most lining fibroblasts can be strongly positive for CD248, only a proportion of SSL fibroblast dissecting through the adipose tissue patches are. Areas of the SSL with dense connective tissue bands are CD248 negative. In RA, IHC confirms retention of CD248^+^ SL and an increase in the CD248^+^ SSL fibroblasts, which is consistent with the significant increase of CD248 in the RA fibrous SSL detected by spatial proteomics significant increase of CD248 in RA fibrous SSL (Figure 6B). Both discoveries aligned with the expression of CD248 in healthy fibroblast-like synoviocytes and stromal cells, and their reported CD248 overexpression in the inflamed synovium of RA patients [38]. This heightened expression situation makes CD248 a marker indicating activated synovial fibroblasts. Another fibroblast marker, Fibroblast activation protein-α (FAP), is linked to RA pathogenesis by contributing to ECM degradation, tissue remodeling, synovial inflammation, and fibroblast activation. Elevated FAP expressions in RA synovium support chronic inflammatory and destructive processes in RA [39,40]. Immunohistochemical detection of FAP^+^ fibroblast was absent in NH synovium but very abundant in the SL and SSL of the RA synovium (Figure 6G,K). The spatial proteomics data also consistently quantified FAP with higher levels in RA SL and fibrous SSL with lowest levels in association with immune cell microenvironments (Figure 6C).

## 4. Discussion

The complexity of the proteome, particularly in the context of a multifaceted disease like RA, and in a histological heterogenous tissue like synovium, remains a significant challenge. RA involves numerous molecular pathways, and the vast array of proteins and their post-translational modifications contribute to the difficulty of fully deconvoluting the disease mechanisms. To address this complexity, it is necessary to employ multiple innovative techniques, such as combining LCM-based proteomics, single-cell RNA sequencing, and mass spectrometry, to capture a more detailed and dynamic picture of the proteomic landscape. These tools enable the identification of distinct proteomic profiles in specific cell types and tissue regions or microenvironments, helping to unravel the molecular interactions at play in RA. This study only scratches the surface, and continued technological advancements are required to further dissect the intricacies of the RA proteome.

Besides the differential and pathway enrichment analysis, the generated LCM proteomics dataset was interpreted mainly in terms of two categories of proteins: membrane proteins and ECM proteins. The membrane proteins play key roles in both RA pathogenesis and treatment as therapeutic targets. They are related to the immune system’s signaling and cellular communication processes that are disrupted in RA. Several known types of membrane proteins involved in RA mechanisms include cytokine receptors, cell adhesion molecules, Toll-like receptors, and immune checkpoint proteins [41,42]. Some existing and emerging therapies inhibiting membrane protein functions have significantly improved RA outcomes by reducing inflammation and modulating immune responses, like TNF inhibitors, IL-6 receptor antagonists, CTLA-4 agonists, and JAK inhibitors [43,44,45]. All of these make membrane proteins to be important therapeutic targets for RA. On the other hand, ECM proteins play a critical role in joint structure and integrity, and their dysregulation is central to the pathological progression of RA. Some known ECM pathological features of RA include cartilage degradation, synovial fibrous thickening, ECM remodeling, and immune activation [46]. ECM proteins are also being targeted or modulated in various therapeutic strategies to mitigate joint damage in RA like matrix metalloproteinase (MMP) inhibitors, TNF-α and IL-1 inhibitors, hyaluronan (hyaluronic acid) therapy, and collagen-based therapeutics [46,47]. Here, we identified 18 up-regulated differentially expressed matrisome proteins in RA compared with NH. They are consistent with multiple targets from previous proteomics reports in RA. CTSZ is reported as one of seven proteins contributing to immune cell infiltration and pannus formation of the RA synovial membrane [48]. Calprotectin (S100A8 and A9) are promising plasma and synovial fluid biomarkers for RA [49], and also up-regulated in the RA synovial membrane based on proteomics [50]. PLOD enzymes are key collagen crosslinkers and contribute to the development of synovial fibrosis in the joints [51]. Analyzing membrane and ECM proteins aids in identifying targets for RA; however, this alone is insufficient to fully explore the comprehensive proteomics dataset generated in this study. The same dataset could further guide new insights by enabling exploration from various perspectives.

To further digest the cellular compositions of distinct histopathological regions in RA synovium, we performed a multi-omics integration analysis combining current spatial proteomics dataset with a published RA synovium scRNA dataset reference [1]. We calculated the region/cell type associations as enrichment z-scores by querying the average cellular expression of up-regulated region-specific gene sets in each cell subcluster to investigate on the cellular composition alternations in histologically defined areas during disease pathogenesis. For example, sublining fibroblast cell types (Figure 5B, F-2 to F-7) were enriched in the RA fibrous SSL region and relatively depleted in the RA SL compared to NH SL and SSL. This could suggest either proliferation or migration of Fibroblast-Like Synoviocyte (FLS) cells from superficial lining to deeper sublining layers or expansion from deeper sublining regions that allow for the repopulation of the RA sublining. Despite its large patient cohort size, one caveat of the current scRNAseq dataset reference is that it lacks single cells from normal healthy donors. A more comprehensive dataset including synovium from healthy controls would further improve the enrichment analysis outcomes.

Similarly, studies with small sample sizes like ours, comprising only three healthy donors and six RA patients, limit the generalizability of our findings, too. Additionally, the absence of paired comparisons between healthy and diseased tissues for the same individual restricted the ability to control individual genetic or environmental factors that might influence the disease process. Although this preliminary study provides valuable insights, the inclusion of larger cohorts with paired samples in future research will be essential to validate these findings and provide a more comprehensive understanding of the proteomic changes in RA. Expanding the sample size will also improve the statistical power and help identify subtle molecular differences that may be critical for understanding RA pathogenesis and progression.

Utilizing FFPE tissue for proteomics analysis continues to pose several challenges, including issues such as protein cross-linking [52], incomplete protein extraction and recovery [53], modification artifacts [54], and limited reproducibility [53]. Ongoing techniques like deparaffinization, rehydration, and high-temperature- and pressure-induced antigen retrieval have been used to help decrosslinking and facilitate protein extraction [4]. These improvements will be particularly advantageous for clinical research of FFPE archives, where fresh or frozen samples might not be accessible. Still, no sample treatments can fully reverse and correct the protein modifications from prior crosslinking with complete protein extraction and recovery, particularly for low-abundance proteins. We should keep it in mind when interpreting the MS results from FFPE samples.

Despite the inherent limitations including chemical modifications, FFPE samples may yield results and proteome coverage comparable to fresh frozen (FF) samples. In a recent study by Erin Humphries and colleagues, a quantitative comparison was conducted on the proteome and phosphoproteome of matched FFPE samples with deparaffinization and FF tissues [55]. Excellent overlap of 85–97% for the proteome and 82–98% for the phosphoproteome were observed between the two preservation methods. The study also demonstrated strong alignment between paired FFPE and FF tissues, as evidenced by Pearson correlation coefficients of 0.93–0.97 for the proteome and 0.79–0.87 for the phosphoproteome. Additionally, an analysis to detect chemical modifications induced by formaldehyde indicated that only 0.05% of precursors were unique to FFPE samples, suggesting limited potential false positives. This significant overlap and correlation underscore the effectiveness of FFPE tissues as a valuable resource for proteomic studies [55,56].

Back to our current study design and experiment, our main aim is to perform proteomic profiling on histopathological regions from RA synovium tissues by laser capture microdissection. H&E staining slice sections from FFPE tissue blocks provide indispensable guidance on targeted region selection. Generally, we cannot attain precise lining and sublining region samples from conventional FF tissue dissection. Moreover, low-abundance protein amounts from tiny LCM samples requires our sample preparation method to be a simple one-pot extraction without downstream clean-ups and transfers. Considering all of these conditions, we need to perform the sample preparations in a strategic and compatible way with LCM samples from FFPE slices. Based on the cutting-edge deep visual proteomics method [4], we processed LCM samples via heating-induced antigen retrieval to remove protein crosslinks. All samples are processed in low liquid volumes and in 96-well plates right after LCM cutting without any sample transfer to reduce sample loss and boost proteome coverage. These methods successfully extract high quality peptides from LCM samples as small as a 0.5 mm^2^ area. Furthermore, in the original paper, the authors used this method to identify more than 5000 proteins from 50 to 100 U2OS cells from FFPE LCM sections [4], strongly supporting the usage of low amounts of FFPE LCM samples for the molecular proteomic profiling. 

The complexity of proteoforms in disease states is primarily driven by alternative splicing and protein post-translational modifications (PTMs). Certain rheumatoid arthritis (RA)-related proteoform signatures may indicate novel pathological cell types or subclusters. Traditional differential analyses of overall protein expression often obscures the modifications arising from these splicing isoforms or PTMs. Nonetheless, the comprehensive analysis and untargeted quantifications of proteoforms pose significant challenges, necessitating precise detection of unique exons or specific PTMs. For example, FnEDA is an isoform of Fibronectin 1 protein containing a unique EDA peptide with 91 amino acids, produced through alternative splicing. FnEDA is typically minimal in human adults, but is upregulated in response to tissue injury, repair, or remodeling [57,58]. FnEDA is expressed in growing tumors, fibrotic pathologies, and autoimmune diseases such as rheumatoid arthritis and inflammatory bowel disease [59,60]. We identified two EDA sequence-specific precursor peptides (GLAFTDVDVDSIK and IAWESPQGQVSR) in the DIA-NN peptide spectral match output report file (report-first-pass.tsv; Appendix A; high Cstore with fidelity; protein group and peptide Q.value < 0.01) and precursor ion quantification output files (report-first-pass.pr_matrix.tsv; Appendix A). For the entire global proteome, the CScores range from highest 1 to lowest 0.83694 among all 405,205 PSMs. Here, by reanalyzing the DIA-NN identification outputs and timsTOF-generated raw instrument files with Skyline software (version 23.1.0.455), we displayed several exemplary spectra (GLA and IAW peptide included) covering the CScores range (0.83694 to 1) for proof of data quality (Appendix A). These results confirm the presence of this disease-associated isoform in human RA synovial membranes. These disease-driving proteoforms could be potential prognostic biomarkers or therapeutic targets. Moreover, protein PTM detections require optimized sample preparations with enrichment as well as high-quality curated spectral library. Top-down untargeted mass spectrometry proteomics can provide more comprehensive protein PTM landscapes. The detection and analysis of these proteoforms including splicing isoforms and PTMs await future technical and computational breakthroughs.

This study also underscores the relevance of proteomic data to the field of genetic medicine, particularly in drug development. Membrane proteins play a critical role in cellular communication, immune signaling, and disease progression, making them highly valuable targets for therapeutic intervention. In RA, for instance, membrane proteins such as cytokine receptors and cell adhesion molecules are central to inflammatory pathways. The insights gained from proteomic analysis of membrane proteins can complement genetic data, leading to the identification of novel drug targets and the development of more effective, personalized treatments. As genetic medicine advances, integrating proteomic data with genomic information will enhance our understanding of disease mechanisms and accelerate the discovery of new therapeutics tailored to individual patient profiles.

## 5. Conclusions

In conclusion, this study utilized the laser capture microdissection technique to enhance our understanding of the complex pathological mechanisms underlying rheumatoid arthritis. By integrating single-cell RNA sequencing data with LCM-based proteomics, we were able to uncover unprecedented insights into the heterogeneity of RA tissues, specifically targeting pathological regions to reveal the intricate cell types and molecular process involved. This study also highlighted the significant roles of extracellular matrix proteins (PLOD2, OGN, and LUM) and membrane proteins (TYROBP, AOC3, SLC16A3, TCIRG1, and NCEH1) in RA pathogenesis, emphasizing their contributions to inflammation, tissue remodeling, and joint destruction. Additionally, key markers (CD90, CD248, and FAP) identified through this analysis were validated using immunohistochemistry, further supported by historical data, reinforcing their potential as therapeutic targets or biomarkers for RA. This study provides a powerful framework for dissecting tissue-specific disease processes and advancing therapeutic strategies in RA.

## Figures and Tables

**Figure 1 proteomes-13-00017-f001:**
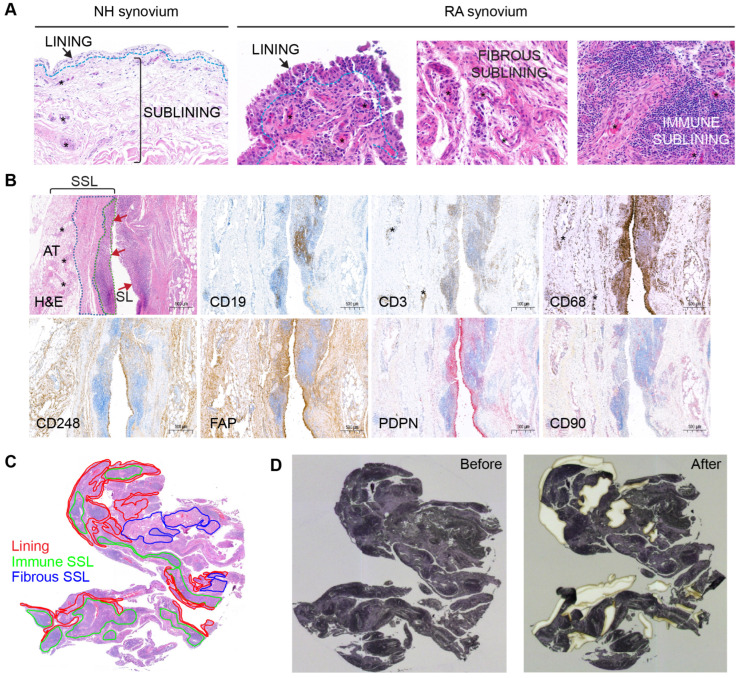
Synovial annotations and tissue dissection. (**A**) Microscopic compartments or microenvironments in NH and RA synovial samples including lining, fibrous sublining and immune sublining. (**B**) Immunohistochemical confirmation of a lymphoid-myeloid RA pathotype, with representation of synovial lining (SL, red arrows) and synovial sublining (SSL). Synovial histopathology based on H&E stain shows that the SSL is divided into immune-rich superficial areas (green dotted line) and fibrous-rich deeper areas (blue dotted line). Immune cell markers for CD19 (B cells), CD3 (T cells), and CD68 (macrophages) confirm that these cells are abundant in the immune-rich areas compared to fibrous rich-areas. By contrast, fibroblast markers like CD248 and FAP predominate outside immune-rich areas. The PDPN^+^ fibroblasts are preferentially distributed in the SL and superficial SSL surrounding the immune cell aggregates. The SL contains CD68^+^macrophages and CD248^+^ FAP^+^ PDPN^+^ and CD90^−^ fibroblasts. The CD90^+^ fibroblasts are only detected in the SSL and predominantly oriented around blood vessels (asterisks). The deepest layer of the SSL is composed of adipose tissue (AT) rich in large blood vessels. (**C**) Pathologist various ROI annotations on the H&E synovium tissue section encompassing 2 mm^2^ from the SL, the immune SSL, and the fibrous SSL. (**D**) Transferring of ROIs to the hematoxylin-stained synovial section on the PEN-membrane slide and laser-capture microdissection of ROIs.

**Figure 2 proteomes-13-00017-f002:**
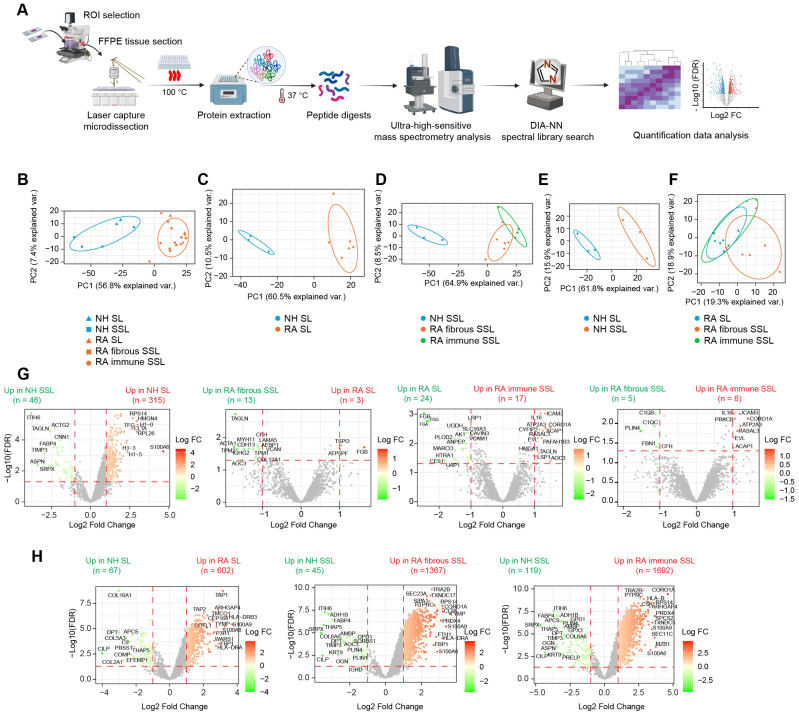
Proteomic profiling of different histopathological regions in NH and RA synovium. (**A**) Proteomics workflow coupling laser capture with LC-MS analysis. (**B**–**F**) PCA plot showing discrimination among disease states and histopathological regions: (**B**) PCA clustering of all 5 region types. (**C**) PCA clustering of lining regions comparing NH versus RA. (**D**) PCA clustering of sublining regions comparing NH sublining, RA fibrous sublining and RA-immune sublining. (**E**) PCA clustering of NH regions comparing lining versus sublining. (**F**) PCA clustering of RA regions comparing RA lining, RA fibrous sublining, and RA-immune sublining. (**G**) Differential expression analysis between different histopathological regions. Horizontal dashed lines represent the adjusted *p* value or FDR of 0.05. Vertical dashed lines represent the fold change of 2 (Log_2_ FC of 1). (**H**) Differential expression analysis between disease conditions. Horizontal dashed lines represent the adjusted *p* value or FDR of 0.05. Vertical dashed lines represent the fold change of 2 (Log_2_ FC of 1). Some representative differential expressed proteins are labeled.

**Figure 3 proteomes-13-00017-f003:**
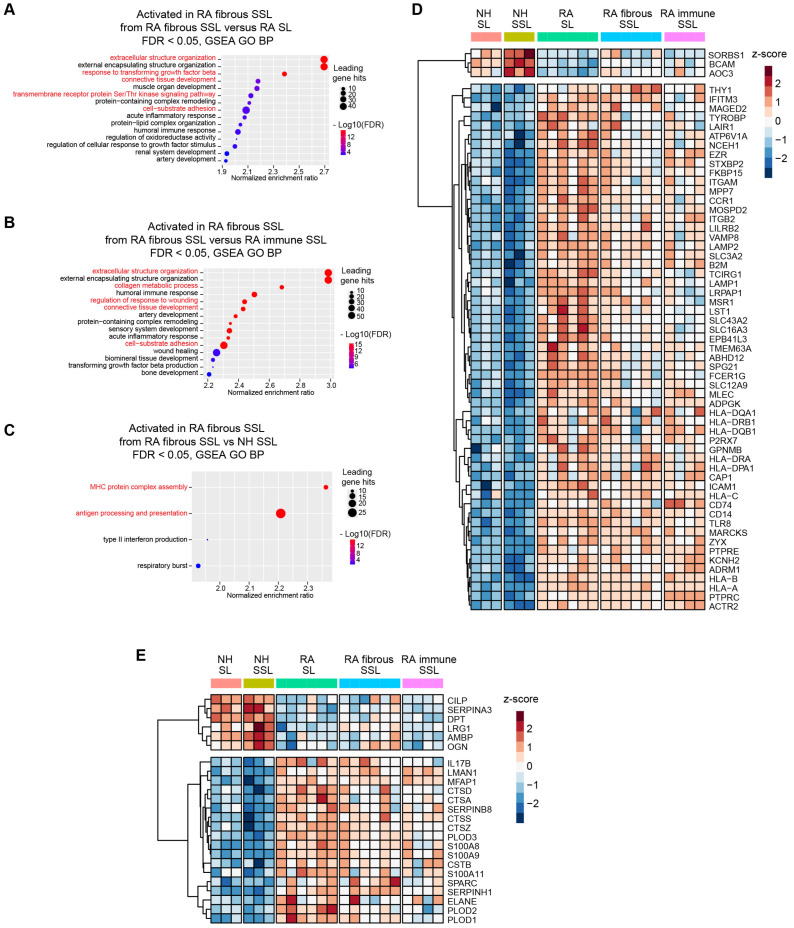
Membrane protein and matrisome protein profiling in different synovium regions. (**A**–**C**) Gene Set Enrichment Analysis (GSEA) against Gene Oncology (GO) terms of Biological Process (BP) using pairwise protein expression comparisons (Log_2_ FC) among different histopathological regions identified enrichment of pathways related to membrane protein processing, signaling pathways and extracellular structure organization, especially in RA fibrous SSL (red highlight). (**A**) RA fibrous SSL versus RA SL. (**B**) RA fibrous SSL versus RA-immune SSL. (**C**) RA fibrous SSL versus NH SSL. (**D**,**E**) Clustered heatmaps of membrane and matrisome proteins that show differential expression between RA and NH groups (appear in both RA SL versus NH SL, and RA SSL versus NH fibrous SSL). (**D**) 57 up-regulated proteins and 3 down-regulated differentially expressed membrane proteins comparing RA versus NH groups. The whole protein expression matrices are normalized by protein rows and clustered based on correlation distance. (**E**) 18 up-regulated proteins and 6 down-regulated differentially expressed matrisome proteins comparing RA versus NH groups. The whole protein expression matrices are normalized by protein rows and clustered based on correlation distance. The sample group columns are not clustered.

**Figure 4 proteomes-13-00017-f004:**
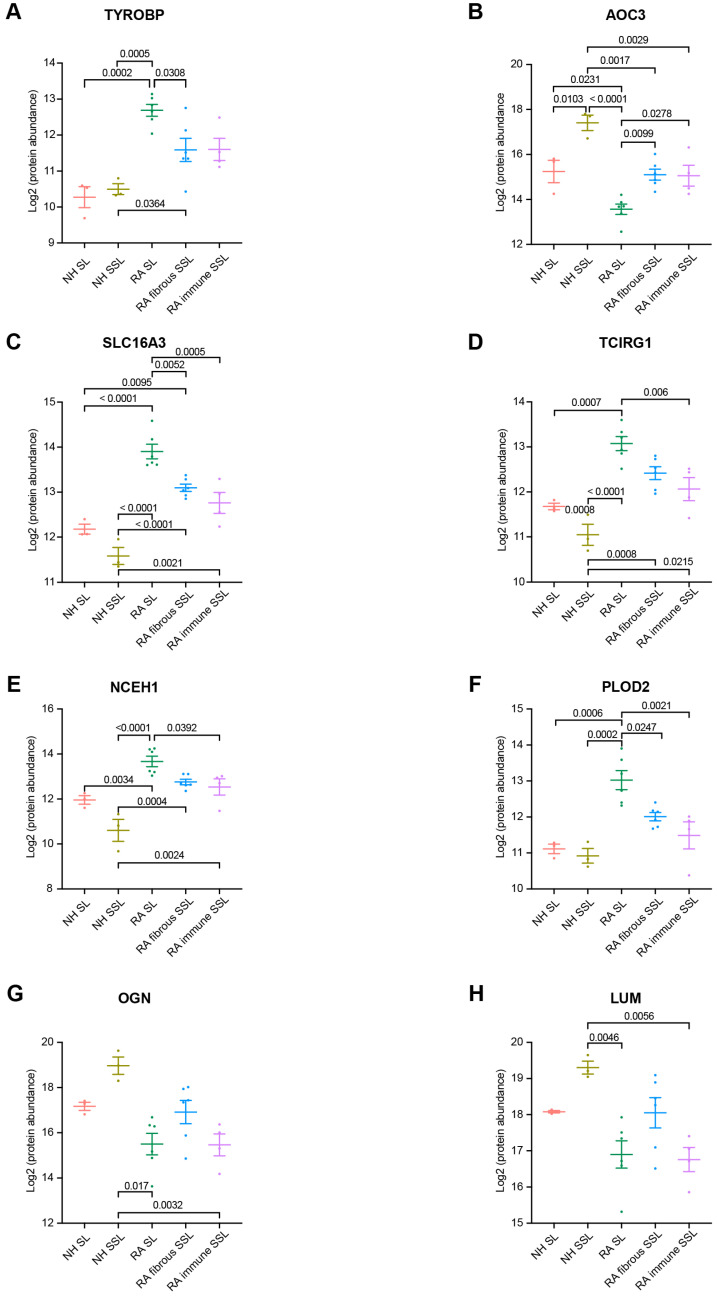
Marker protein expression comparisons among region types. (**A**) TYROBP. (**B**) AOC3. (**C**) SLC16A3. (**D**) TCIRG1. (**E**) NCEH1. (**F**) PLOD2. (**G**) OGN. (**H**) LUM. We focus on all 10 pairwise comparisons from 5 groups. The numbers above the square brackets indicate *p* values < 0.05 from one-way ANOVA followed by Tukey’s multiple comparison tests.

**Figure 5 proteomes-13-00017-f005:**
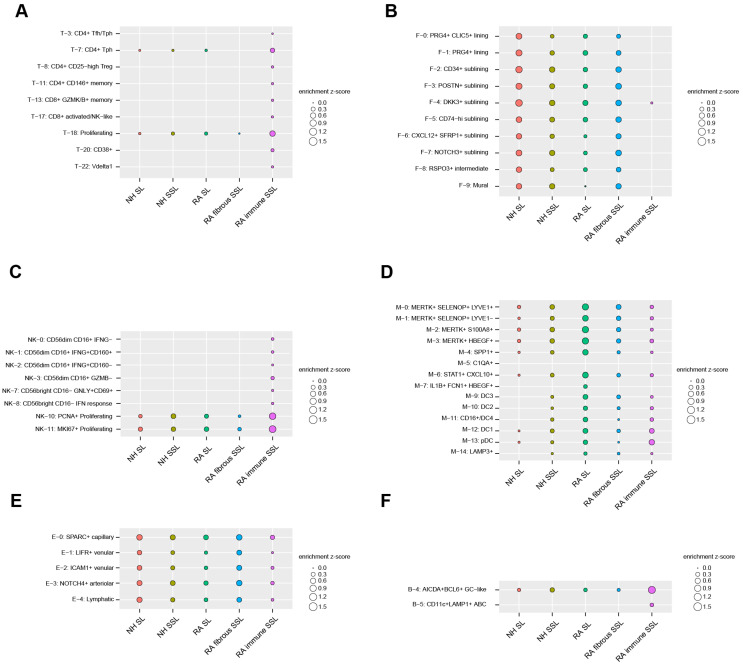
Cell subcluster enrichments in different histopathological regions by the Query method integrating LCM spatial proteomics and scRNA-seq transcriptomics. For clarification, we only plot subclusters that display enrichment in at least one of five region types. All of the depleted subclusters are illustrated in Appendix A. (**A**) 8 T cell subclusters. (**B**) 10 stromal cell subclusters (fibroblast and mural cells). (**C**) 8 NK cell subclusters. (**D**) 14 myeloid cell subclusters. (**E**) 5 endothelial cell subclusters. (**F**) 2 B/plasma cell subclusters.

**Figure 6 proteomes-13-00017-f006:**
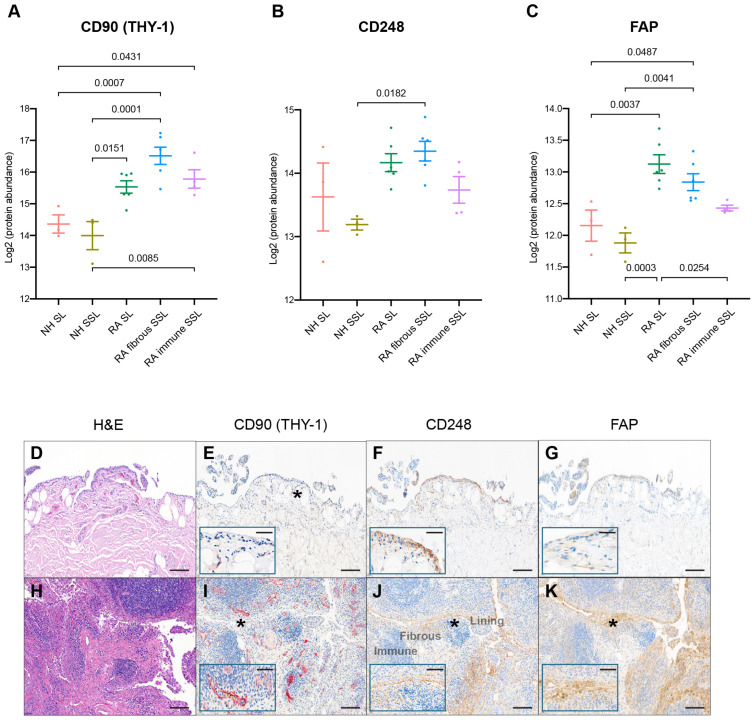
IHC comparison with spatial proteomics analysis. The protein expression of fibroblast markers like (**A**) CD90, (**B**) CD248, and (**C**) FAP with LCM proteomics were plotted. The numbers above the square brackets indicate *p* values < 0.05 from pairwise differential expression analyses. The immunohistochemical characterization of healthy (**D**–**G**) and RA (**H**–**K**) human synovium confirms that healthy fibroblasts are CD248^+^ CD90^−^ and FAP^−^. The scale bar is 50 µm. The expressions of CD90 (**E**,**I**), CD248 (**F**,**J**), and FAP (**G**,**K**) are increased in fibroblasts of the RA synovium, while SL fibroblasts remain CD90^−^. The expression of CD90 is increased in RA, although the perivascular and vascular localization of CD90^+^ fibroblasts is maintained in health and disease (inset). H&E in (**D**,**H**); the insets in (**E**–**G**,**I**–**K**) show higher magnification (*). Synovial lining, fibrous sublining, and lymphoid aggregate in immune sublining, as labeled. The scale bar is 10 µm in insets.

**Table 1 proteomes-13-00017-t001:** Demographic information of subjects on the sex, age, diagnosis, and joint site sampled.

Subject ID	Disease Status	Race	Age (years)	Sex (M/F)	Joint Site	Pathotype
NH1	Healthy	Caucasian	55	M	Knee	Non-lesional
NH2	Healthy	Caucasian	96	F	knee	Non-lesional
NH3	Healthy	Caucasian	75	F	Knee	Non-lesional
RA1	RA	Caucasian	31	F	Biopsy	Lymphomyeloid
RA2	RA	Unknown	61	F	Hand	Diffuse Myeloid
RA3	RA	Unknown	75	F	Hand	Lymphomyeloid
RA4	RA	Unknown	68	F	Biopsy	Lymphomyeloid
RA5	RA	Caucasian	57	M	Knee	Lymphomyeloid
RA6	RA	Caucasian	60	M	Hip	Lymphomyeloid with TLF-like structures

## Data Availability

The mass spectrometry proteomics raw files and DIA-NN search output files have been deposited to the ProteomeXchange Consortium via the PRIDE partner repository (https://www.ebi.ac.uk/pride/, accessed on 7 April 2025) with the dataset identifier PXD062696.

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
