# Peer review of "Integrative Spatial Proteomics and Single-Cell RNA Sequencing Unveil Molecular Complexity in Rheumatoid Arthritis for Novel Therapeutic Targeting"

_proteomes, 2025, doi:10.3390/proteomes13020017_

Round 1
Reviewer 1 Report
Comments and Suggestions for Authors
Conclusion:
This study utilized the laser capture microdissection technique to enhance our understanding of the complex pathological mechanisms underlying rheumatoid arthritis. By integrating single-cell RNA sequencing data with LCM-based proteomics, we were able to uncover unprecedented insights into the heterogeneity of RA tissues, specifically targeting pathological regions to reveal the intricate cell types and molecular process involved. By combining high-resolution spatial proteomics and transcriptomic analyses, the authors provided novel insights into the molecular mechanisms driving RA and highlight potential protein targets for therapeutic intervention.
Comments
1. Please highlight the number of the differential expression proteins in Fig2G-2H.
2. In Figure 2F, the clustering analysis shows that the RA SL, RA fibrous SSL and RA immune SSL groups cannot be clearly distinguished. In Figure 2G, many differentially expressed proteins were identified through proteomics. Please provide a detailed explanation for this.
Reviewer 2 Report
Comments and Suggestions for Authors
This study entitled “Integrative Spatial Proteomics and Single-Cell RNA Sequencing Unveil Molecular Complexity in Rheumatoid Arthritis for Novel Therapeutic Targeting” combines laser capture microdissection (LCM)-based spatial proteomics and single-cell RNA sequencing (scRNA-seq) to investigate synovial tissue heterogeneity in rheumatoid arthritis (RA). By analyzing diseased and healthy synovial lining/sublining regions, the authors identified spatially dysregulated membrane proteins (e.g., TYROBP, AOC3) and ECM proteins (PLOD2, OGN), while scRNA-seq integration revealed enriched immune/stromal cell populations. Despite the innovative multi-omics approach, several questions need to be answered and some improvements are needed:
The average age of NH control group is significantly older than that of RA patients, introducing potential age-related bias. The rationale for this age disparity and its impact on synovial protein/cellular profiles needs clarification.
Why not MALDI-MSI instead of LCM-MS? It is not fully discussed what advantage LCM-MS gives over MALDI-MSI in introduction section for this study.
Could author elaborate why reduction/alkylation steps are skipped in the sample prep before digestion?
The spectrum library in-silico generated from DIA-NN, does it include the predicted spectrums for modified peptides? As matrisome proteins are heavily modified (lysine/proline hydroxylation etc.,), the protein quant result searched by a spectral library excluding these abundant PTMs might lead to biased identifications on matrisome proteins, especially PTMs is mentioned in the discussion section as one factor contributing to the complexity of proteoforms in disease states.
In 2.6 integration of LCM proteomics and scRNA dataset, “For each gene with p < 10−2, we examined which region the effect size was highest and determined that protein to be specific or uniquely up-regulated for this region”, is there any reference supporting such method to decide upregulated region-specific proteins?
Figure 2B, did author look into the PCA loadings? as loadings could give information on which group of proteins make highest contribution to separate on each PC. Such list could be cross compared with differential expression analysis in figure 2 G/H.
24 matrisome proteins are mentioned as significantly changed in RA samples. Again, if the differentially expressed matrisome proteins are originated from searching against a spectral library without ECM-specific PTMs, some differentially expressed ECM proteins might be overlooked.
Did author compare the findings of upregulated/downregulated matrisome proteins in this study with any comprehensive ECM proteomics database that includes ECM protein quant result across healthy and diseased tissue, to see the agreement? e.g., MatrisomeDB
In figure 4, are the 8 proteins selected randomly from the upregulated/downregulated protein pool?
The depth of multi-omics integration needs to further improve, as now it limits to cell-type enrichment analysis only, maybe additional protein-transcript correlation map would help strengthen the conclusions.
In figure S3, how does the peptide ID number or sequence coverage look like between LCMs with varied area sizes?
I didn’t see the DIA-NN search result table files in supplementary files (protein/peptide IDs and original protein quant result), will these be attached to PRIDE repository with raw files?
Reviewer 3 Report
Comments and Suggestions for Authors
The manuscript was very well written and edited. It describes results of a study using laser-captured tissue from formalin-fixed paraffin-embedded blocks for proteomic analysis. While the authors discuss their limitation of too few samples, they do not consider any of the technical limitations of their method.
First, using FFPE-tissue for proteomics is highly debated, because the formalin treatment affects the proteins and introduces multiple, in part unknown, modifications. Those affect database search and protein assignment and may cause false-positives.
Second, when discussing certain proteoforms of interest, the authors should not only mention the peptide sequences they use for the assignment, but also show the respective mass spectra proving the assignment. It is not rare that spectral quality is so low that the assignment cannot be confirmed by the human operator. While the TimsTOF is an excellent instrument, it is not guaranteed that all data come out equally well, in fact, they will not.
Thus, the authors must discuss these obvious technical limitations and provide proof of the data quality they are obtaining.
Round 2
Reviewer 2 Report
Comments and Suggestions for Authors
Thanks authors for addressing the comments.
One little point: Figure S3 doesn't seem to have a title.
Author Response
Comment 1: One little point: Figure S3 doesn't seem to have a title.
Response 1: Thank you for the constructive suggestion. We added the title “Proteins with maximum PC1 and PC2 loadings from PCA in 7 pairwise region comparisons overlap with the differentially expressed proteins” in Figure S3. In the final publication, all titles will subsequently appear in the corresponding figure legend.
Reviewer 3 Report
Comments and Suggestions for Authors
While the authors have improved their manuscript, the main criticism has not been addressed.
FFPE tissue samples contain proteins which have been modified in several different ways by the treatment. These modifications influence the proteomics experiment as they change the mass of the peptides. It is not possible to predict all modifications which might occur, nor is it advisable to unduely enlarge the search space by programming them all into the algorithm, because this increases the number of false positive hits. I believe that the authors are not aware of the problem, because they argued that they are not interested in the modifications and thus do not see a need to pay attention to them. The problem is that the sample proteins are not well defined due to prior treatment (garbage in) and that they cannot be properly analyzed with wrong expectations (parameters of the search algorithm – garbage out). This effect will likely not be a problem with abundant proteins, because sufficient unmodfied peptides can then be detected, but for low abundant proteins with few detected peptides, mismatches will increase.
The authors state that they controlled the protein quality and it was good. What does this mean? How do you control the quality of the proteins? You can measure their concentration and possibly run them on a gel, but what was their criterion of “good quality”?
As for the proof of the quality of their data in general, they still did not show any spectra. Peptide fragment ion spectra are the ultimate proof for data quality – if they are good (sensible signal/noise ratio, peaks with isotopes showing, ion series visible above parent ion in the m/z range), results of software algorithms tend to be reliable. Unfortunately, especially at low concentrations, spectral quality is often not sufficient leading to questionable assignments the algorithm comes up with under the cricumstances. Thus, it is necessary to show a few spectra of important peptides and demonstrate their quality. This is something every analytical chemist working in this area can evaluate in contrast to software scores, because the differ across platforms and instruments. It would be helpful to show a peptide fragment ion spectrum of top quality with its score and also show a spectrum for the lowest score they are using; even better: show 3-5 spectral examples for the range of scores they are considering.
Moreover, there are a number of critical papers available concerning FFPE – omics. They should be taken into account.
Round 3
Reviewer 3 Report
Comments and Suggestions for Authors
Thank you for providing some spectral output if only in the Supplement. It appears to be from Skyline, which transforms original profile data into centroided spectra. It is often useful to really look at the original profile data to evaluate data quality, because software can always misinterpret. The choice of presentation is acceptable here, but the legend must be corrected. You show both chromatgrams and centroid spectra, which is not the same coming from two different instruments. Moreover, the reader should be informed that the output is taken from Skyline analysis. Also, how does Skyline analysis relate to your data mining? Nowhere in the paper it is mentioned.
Furthermore, no sample treatment, high pressure or not, is capable of reversing fully and correctly the damage done to proteins done by prior crosslinking. The limitations of the analysis should be stated more realisticly.
Author Response
Comment 1: Thank you for providing some spectral output if only in the Supplement. It appears to be from Skyline, which transforms original profile data into centroided spectra. It is often useful to really look at the original profile data to evaluate data quality, because software can always misinterpret. The choice of presentation is acceptable here, but the legend must be corrected. You show both chromatograms and centroid spectra, which is not the same coming from two different instruments. Moreover, the reader should be informed that the output is taken from Skyline analysis. Also, how does Skyline analysis relate to your data mining? Nowhere in the paper it is mentioned.
Response 1: We add more details of spectral output legend and speak out the figures are coming from Skyline analysis in the discussion part. In addition, we added one section in method part to articulate the procedure of our Skyline analysis. The MS1/MS2 chromatograms and MS2 centroid spectra are both from Skyline analysis using the raw Bruker timsTOF instrument spectra files with DIA-NN spectral library and output identification file (report-first-pass.tsv). All these files are provided in the ProteomeXchange Consortium via the PRIDE partner repository with the dataset identifier PXD062696. The Skyline analysis is used for spectral quality control and our main data mining and analysis are based on the quantitative protein group intensity output from DIA-NN.
Comment 2: Furthermore, no sample treatment, high pressure or not, is capable of reversing fully and correctly the damage done to proteins done by prior crosslinking. The limitations of the analysis should be stated more realistically.
Response 2: Thank you for pointing that out. We are aware that we cannot fully reverse these modifications to restore the naive protein condition. We have revised our manuscript to more clearly and realistically state these limitations.
Round 4
Reviewer 3 Report
Comments and Suggestions for Authors.